# Elucidating Collapse-Resistant Mechanisms of Pore Geometries in Fire Ant Nest Cavities

**DOI:** 10.3390/biomimetics9120735

**Published:** 2024-12-03

**Authors:** Tyler Felgenhauer, Satchi Venkataraman, Ethan Mullen

**Affiliations:** 1Department of Aerospace Engineering, San Diego State University, San Diego, CA 92115, USA; satchi@sdsu.edu; 2Department of Computer Science, San Diego State University, San Diego, CA 92115, USA; emullen9665@sdsu.edu

**Keywords:** bio-inspired design, fire ant nest cavities, porous structures, collapse resistance, hierarchical structures

## Abstract

Porous materials and structures, such as subterranean fire ant nests, are abundant in nature. It is hypothesized that these structures likely have evolved biological adaptations that enhance their collapse resistance. This research aims to elucidate the collapse-resistant mechanisms of pore geometries in fire ant nests. Finite Element Models of ant nests in soil were generated using X-ray CT imaging of aluminum castings of ant nests. Representative volume elements of the ant nests, representing porous structures at various depths, were analyzed under confined compression. This work on investigating fire ant (sp. Solenopsis Invicta) nests found them to be hierarchical and graded at various depths that affect how they resist loads and collapse. The top portion acts as a protective shield by distributing damage and absorbing energy. In contrast, the lower chambers localize stress, contributing to damage tolerance. This research provides evidence to suggest that ant nests have developed properties that allow them to resist collapse. These findings could inform the design of lightweight and durable cellular structures in various engineering fields.

## 1. Introduction

Porous materials have a wide variety of applications in engineering design. Porous materials with interconnected networks of cells or pores are commonly referred to as cellular materials. Their cells can be ordered (architected) or random (stochastic). They can be open cell, where the pores are interconnected and provide free passageway for fluids, or closed cell, where each cell is isolated and encloses a volume within. Cellular materials are lightweight and have high stiffness to weight and strength-to-weight ratio, and therefore, are used as components in lightweight structural design and construction [1].

Cellular materials such as honeycomb sandwich cores are utilized in composite structures due to their exceptional strength-to-weight ratio, high stiffness, and excellent energy absorption properties [2,3]. Metal and polymer foams are widely used due to their high strength-to-weight ratios and energy absorption capabilities [1,4]. Their large internal surface-to-volume ratio leads to slender internal structures that can undergo extreme deformation and absorb energy, which makes them useful as impact energy absorption and thermal insulation applications [4,5,6]. Chen et al. [7] emphasized that the combination of high thermal stability, low thermal conductivity, and reduced weight makes porous ceramics a promising material choice for various high-temperature and acoustic applications.

Advances in additive manufacturing allow for the creation of complex porous geometries that were previously impossible to achieve. These include lattice structures and intricate pore networks that optimize mechanical and thermal properties. Halbig [8] found that additive manufacturing technologies offer significant advantages, including enhanced design flexibility, reduced material waste, and the ability to produce complex geometries that are not possible with traditional manufacturing methods. While additive manufacturing enables the creation of complex porous structures, it faces challenges such as material limitations, resolution constraints, high costs, post-processing requirements, and scaling difficulties that can impact its widespread adoption [9].

Cellular materials are required to be open celled or vented for use in space structures, as under vacuum, closed-cell foams can be damaged. Vented cellular materials generally have lower stiffness and strength than closed-cell structures [10]. This loss in performance in the current state of the art prompts the need to develop more efficient and stronger vented cellular structures to address the research gap.

However, nature offers a wealth of inspiration for overcoming challenges. Throughout human history, nature has often been a source of inspiration [11,12], a process known as biomimetic design. Natural systems have profoundly influenced the development of advanced technologies in aerospace engineering, such as flexible wings that adjust shape in response to airflow based on birds [13], riblets inspired by shark skin to reduce drag [14], and materials that replicate the shock-absorbing qualities of spider silk [15]. Breish et al. [16] examined six essential strategies developed in nature for efficient mechanical load management: hierarchical composite structures, cellular architectures, functional material gradients, hard-shell soft-core designs, form-following-function principles, and robust geometric configurations. Zhang et al. [17] discussed lightweight structural biomaterials and examined strategies to enhance their mechanical performance and adaptability, drawing on natural systems to create strong, efficient materials for engineering applications.

The design of porous structures has taken much inspiration from the vast number of porous structures that exist in nature such as the microstructures of bone, cork, and wood [18,19]. These structures exhibit desirable mechanical properties such as high bending strength, rigidity, and durability [20]. Chen et al. [21] reviewed methods for preparing biomimetic porous structures, highlighting the effects of micropores, mesopores, and macropores, while evaluating the advantages and disadvantages of hierarchical structures and their preparation techniques, and proposing modification strategies. Examples of natural porous structures used in biomimicry applications include honeycomb for composite applications [5], bone for optimized strength/weight ratios [22], sea sponge for damage-tolerant lattice structures [23], tougher materials inspired by nacre for self-healing materials [24] and self-cleaning surfaces [24], and nano-architecture surfaces for use for reversible adhesion [24]. There is extensive documentation of researchers utilizing biomimicry to design porous structures for specific applications [24,25,26,27,28,29]. Recent advancements, particularly in modeling methods to investigate natural phenomena, have been made in the fields of material and structural design [28,29]. With the major developments in additive manufacturing techniques, replicating some of the more complex porous networks is now a reality [2,30].

Among these complex and functional natural designs, the study of ant nest architectures provides a fascinating example. The study of ant nest architectures [31,32] has revealed that the underground excavated dwellings that ants create can have complex shapes that are adapted to the soil, climate, and habitat they are found in. These nests are vented porous structures, as ants need to create passageways to take excavated materials out of the nests and bring food into the nests. Ants build subterranean nests by excavating cavities that are varied in shapes, forms, and architectural complexity, vented to the outside, and vary by species and habitat.

Studies have used techniques such as excavation and CT scanning to map the intricate tunnel systems of fire ant nests. These studies provide detailed insights into the size, complexity, and organization of nests [31]. Tschinkel’s [31,32] encyclopedic books on ants identified common trends within the nest architecture of certain species, such as that of fire ants. The main features are that porous volume density decreases with depth, porous chamber sizes increase proportionally with depth, and chamber shapes become more complex and lobed with depth. The underground nest consists of a series of interconnected tunnels and chambers. Chambers are often used for brood care, food storage, and waste management. The design of these tunnels and chambers reflects the need for efficient space utilization and environmental control [33]. Buarque de Macedo et al. [34] observed ant tunneling under X-ray CT to investigate how ant nest structures grow. It was found that the arching of soil grains is the mechanism that provides structural stability during excavations. According to Belachew et al. [35], helical shafts are often employed in ant nests, as they work in conjunction with soil arching to provide increased stability. This is accomplished through reducing the risk of tension and shear failure at the cavity walls.

Although ants have been studied extensively for their social dynamics and self-organization [36], few studies have focused on the structural or other engineering properties of ant nest cavities. Qu et al. [37] performed a structural analysis of a single chamber of an ant nest to understand its structural properties. Yang et al. [38] also utilized finite element methods to analyze ant nest geometries to investigate the ventilation properties of the Camponotus japonicus Mayr nest. Understanding the structural adaptations requires modeling the entire nest and investigating their collapse dynamics.

It is hypothesized that these subterranean nest architectures have developed desirable shapes that resist collapse under mechanical loads, providing increased strength and stability, and minimal need for repair. Ants that develop collapse-resistant nests can minimize energy spent in the re-excavation and repair of collapsing nests, and instead direct it to gathering food or expanding the nest to assist in colony growth. For this research, the species of fire ants, Solenopsis invicta, will be the primary focus since fire ants build nest cavities with complex topology [39] (Figure 1), with high levels of porosity that would be desired in the design of porous structures. Investigating whether ant nests have evolved structures that provide desirable collapse resistance and understanding the design features and mechanisms that imbue ant nests with collapse resistance has the potential for the future biomimetic design of vented porous structures. The paper presents a numerical investigation into the collapse resistance of fire ant nests and contributes to new knowledge on the structural properties of fire ant nest cavities. This study uniquely applies fire ant nest architectures to engineered structures, a largely unexplored area of biomimetic design, to address the loss performance in vented porous structures.

## 2. Materials and Methods

This section presents the procedures used to develop models of fire ant nests in soil medium based on X-Ray CT scans (General Electric, USA) of ant nest architectures captured by aluminum casting, the finite element analyses performed to investigate the collapse resistance of fire ant nests, and the methods utilized to analyze the elastic and plastic responses exhibited by the fire ant nest geometries under mechanical loading.

### 2.1. Finite Element Geometry Creation

This research is focused on fire ant nests (Solenopsis Invicta) that are found commonly in the southeast United States in loamy soil [32]. This species of ants was chosen as their nests exhibit a higher level of porosity than the nests of other ant species, which would be desirable in the design of porous engineering structures. Another reason for the selection of the fire ant nests is the observable hierarchical geometry seen in the nests’ architecture. Procedures for studying the physical architectures of ant nests using liquid aluminum castings have been developed. In this study, nest castings, used for the development of finite element models, were obtained by special order from David Gatlin [40].

The castings are created by first inserting a center tube in the center of the ant nest mound for pouring the molten aluminum (at ~1300 °K [1400 °F]) into the cavity. The gravity-assisted flow of the highly molten aluminum (well above its melting temperature of 934 °K [1221 °F]) flows down the cavities and fills the intricate chambers before solidifying. As in any blind castings, there is some uncertainty whether the molten aluminum has penetrated deep enough into all the cavities. Once solidified, the casting with solid is dugout and then carefully cleaned using small water jets to remove any soil, gravel, and vegetation that remain in it. As ant nests have naturally highly stochastic porous structures, a great deal of variability can be seen in the geometries exhibited in different nests, thus making it difficult to establish generalizations in regard to their geometry. In this research, three different ant nest castings were obtained, labeled as Cast 107, Cast 062, and Cast 0P1. Images of these ant nest castings are shown in Figure 1. This work will focus solely on the geometry of Cast 107, as it appears to be a complete nest and its geometry reflects the architectural trends reported in the literature.

These aluminum castings were utilized to construct finite element models capturing the geometry of real-life ant nests. The first step in converting these castings to digital geometry involved imaging using X-Ray Computed Tomography, commonly referred to as CT scans. X-ray CT imaging is a technique that combines multiple X-ray measurements taken from different angles to create detailed cross-sections of an object. By using image-processing algorithms, one can reconstruct the three-dimensional (3D) internal structure with high definition. The resolution of the imaging is a function of the size of the object imaged, the sensitivity of the detector, the number of pixels on the photo sensor, and the number of projections used. This technology is widely used in various fields for non-invasive examination and analysis.

In this work, all X-Ray CT imaging was performed using X-Ray CT equipment (Discovery CT750 HD and GE Lightspeed VCT64 systems), which is used for medical imaging at a commercial medical imaging facility, shown in Figure 1. The voxel resolution of the imaging was 1.25 mm, which was sufficient to capture the fine details present in the casting. The X-ray imaging data from the X-Ray CT, stored as DICOM files, were transferred to an external visualization software for image processing.

The 3-D Slicer™ [41,42] open-source software was used for processing the DICOM files and visualization of the ant nest geometries that were generated from the X-Ray CT. This 3-D Slicer software is used for visualization, processing, segmentation, registration, and analysis, and is often employed in the biomedical/medical fields. It was utilized due to pre-existing expertise operating the software. Here, the 3-D Slicer (Version 5.0.2) was used primarily to convert the data gathered from the X-Ray CT scans (DICOM files) of the ant nest geometries to graphic images (.stl files). The first cleanup step was to eliminate any extraneous components in the image such as the table or support structure that was used to hold the ant nest during the scan.

The next step is to define the boundaries of the ant nest cavities using image-processing algorithms in the 3-D Slicer. The key parameters in this step of the digital reconstruction are the grayscale thresholding and the smoothing operations. The grayscale thresholding technique uses the grayscale value of the images to demark the boundaries of the different components. In this case, it is used primarily to define the aluminum casting surface boundary with respect to air. The small embedded soil particles and internal reflections of the X-rays create grayscale values in some regions. This leads to non-binary values of the gray scale. By using thresholding and smoothing, these artifacts can be removed, and the interfaces can be clearly identified.

The smoothing parameter employed in the first stage of the image processing also helps eliminate sharp corners in the images. These sharp features, if not eliminated, create problems when performing finite element analyses of scanned structures due to the excessive stress concentrations they introduce and need for very fine meshing to mesh such features to avoid excessive distortion of the mesh during analysis. Of the available smoothing operations in the software, the closing smoothing method was utilized. This operation fills sharp corners and holes smaller than the specified kernel size, which was chosen as 2.5 mm for the purpose of this research. The smoothing kernel size was selected following a sensitivity analysis, which is further discussed in previous work [43]. This method does not remove any material from the cast segments of ant nest cavities. Thus, it allows for smoothing without distorting the geometry. The parameters used in the image processing are presented in Table 1. Upon completion of the two image-processing operations, the models were then exported as .stl image files. The .stl files provide a description of the surface.

Fusion 360™ [44] software, developed by AutoCAD^®^, was used in converting graphic image files to solid body files. The models are converted to solids using the default processes provided in the software and exported as a solid geometry that can be read in the finite element analysis (Abaqus™, v2021) [45] software’s preprocessor. For the conversion, there are two applicable file types that are accessible for the usage of Abaqus™: .step and .sat files. It was experienced during this research that the .sat files translate more efficiently to Abaqus™, as they are the native kernel for Abaqus™, while .step files are more effective for maintaining assembly hierarchy. As these models do not deal with complex multilevel assemblies, the .sat files were employed.

### 2.2. Characterization of Ant Nest Geometry

Before conducting the finite element simulations, it is necessary to understand the complex geometrical trends exhibited in the fire ant nest cavities. As the porous network is extremely dense in some sections, it is difficult to visually observe. To accomplish this, skeletonization was used to characterize the porous network of the fire ant nest. Skeletonization is a visualization procedure that allows for the simplified representation of complex porous networks. The skeletonization was performed using an open-source visualization software developed for vasculature datasets known as VesselVio [46]. The visualization software develops a 3D graph network description of the porous cavity structures along with the radii, volume, and length of the pore networks between each branching.

Figure 2 presents the skeletonized representation of the ant nest pore cavities from Cast 107. The skeletonization algorithm was used to determine various properties by taking the average value between branching points. These properties are illustrated in the above figure and are color coded by the pore radii, volume, and lengths between branching points. The skeletonization images show that the radii of the ant nest chambers increase with depth. The volume of the cavities follows the same trend. The bottom chambers have significantly more volume than those at the top. The general observation of the skeleton structure indicates that the ant nest shows distinctively different architectures at the top, bottom, and middle portions (although these sections’ start and end points can be subjective). Figure 2c shows that most chambers at the top portion of the ant nest are horizontally oriented and are connected vertically to their nearest neighbor chamber. The porous network in this region is also much denser than in the rest of the nest. The porous chambers in the middle portion of the nest are primarily oriented vertically. The spacing in between pores (regions of soil) also increases with depth. Figure 1 shows that the pore chambers at the bottom of the nest are lobed. This correlates with the skeletonization images, as there is a sharp increase in volume in these sections. Due to these distinctive features, moving forward, the nest geometries will be classified into top, middle, and bottom portions. It should be noted that due to the stochastic nature of the geometry and the lack of visibility for the inner chambers of the nest, the boundaries of the defined sections are not distinct. Rather, the pore geometry at the boundaries is representative of that from both sections it is separating. The features observed in the skeletonization agree with the descriptions of ant nest architectures in the literature.

### 2.3. Representative Volume Elements

Preliminary analyses utilizing linear-elastic simulations of full ant nests that were placed in a cylindrical soil volume and placed under semi-confined compression tests indicated that the distinct sections serve a global task to shield the bottom central pore in which the queen resides [43]. It was inferred that this behavior occurs due to the horizontally oriented cavities at the top, leading to less stiff regions that, even before collapse, leads stress redistributions laterally away from the ant nest core/center. Further non-linear analyses of the end point segments of the nest geometry, indicative of the global architectural trends exhibited in the ant nest, reveal that these segmented end geometries of pore structures in fire ant nests exhibit superior resistance to collapse when compared to simpler pore geometries with comparable volume fractions due to ant nest cavity geometry causing favorable load redistribution [43]. This indicates that the geometry at various levels can have special functions and contribute to the collapse resistance of the overall nest. However, to investigate this further, the full nest behavior must be understood, which is unobtainable with current tools due to computational challenges.

Understanding even the collapse analyses of the individual distinct cavity shapes and pore structures of fire ant nests can provide insight into the overall collapse of the ant nest. For this, small rectangular volumes of the soil with ant nest cavities were extracted from the three main sections that were identified in the previous section using skeletonization tools, which are shown in Figure 3. These samples were chosen to be sufficiently large enough to capture the behavior of the ant nest cavity structures at the locations from which they were extracted. These sub-volumes are herein referred to as representative volume elements (RVEs). The use of representative volume elements (RVEs) is well-established in the micromechanics of composite materials. This research used nine RVEs in the analyses, with three RVEs each taken at a different depth of the nest. These RVE models are analyzed under confined compression. The RVEs were created by merging solid cubes with a location on the full Cast 107 geometry.

A Boolean subtraction was performed to create a cavity shaped like the pore geometry from the region it was taken. It was decided that the RVEs would be rectangular prism-shaped cubes due to the requirements of a micromechanics plugin used for the homogenization of elastic properties, which are discussed later in this section. To assess the validity of the sizing selection of the RVEs, it must be proven that the behavior exhibited from that RVE is indicative of the response when analyzing the section as a whole. In order to verify our sizing assumptions, multiple additional RVEs were taken from different locations in each section in order to verify that the response was similar to that of the simulations presented. In addition, an RVE with a double aspect ratio was also taken in the top section to further investigate the effect of the RVEs’ size on test results. This proved to be computationally demanding and, therefore, was not repeated for the other sections. In addition to verifying the RVEs’ size as equivalent to the aggregate response, this study was also used to investigate the variations of porous geometry and density in the three defined sections. The RVEs and the locations from which they were derived are shown in Figure 4.

The sizing of the RVEs was selected as to have comparable pore volume fractions. This required different sizing for different sections due to the sharp variation in pore density that is seen through the depth of the nest. The top has a higher pore volume fraction than the bottom and middle due to the distinct cone shape of the overall nest architecture. The sizes were selected as to obtain about 30–40% pore volume fraction per RVE. Since the pore size and spacing vary throughout the fire ant nest, the RVEs taken from each section had to have variations in dimension to ensure that they had comparable pore volume fractions. The sizing of the RVEs per section are top—42.5 × 42.5 mm, middle—35 × 35 mm, and bottom—30 × 30 mm. A sensitivity study was performed to verify that the size of the RVEs did not significantly impact their loading response [43].

The pore volume fraction of the RVEs is calculated by dividing the volume of the negative pore geometries by the volume of the solid cube. This is represented in Equation (1), and the resulting value of pore volume fractions for all nine RVEs are reported in Table 2.
(1)$$V_{f}=V_{pore}/V_{cube}$$

### 2.4. Soil Model Material Properties

This section describes the selection of a constitutive model for the analyses [47,48]. Extensive literature exists on the modeling of soil plasticity with finite element methods and validation of models with experimental results [49,50,51]. Naderi-Boldaji et al. [52] showed that the Drucker–Prager predicted the responses observed in the experiments of soil compression. The study sought to accurately simulate various confined compression tests of soils.

The commercial finite element software Abaqus™ has many built-in constitutive models for soil analysis problems. They include geostatic and soil solution steps that account for gravity/pore pressure loads as well as the response of porous media saturated with fluid [45]. They also allow for the specification of parameters for void ratio, saturation, and pore pressure [45].

For a simplistic representation, the elastic behavior of soil can be represented by an isotropic elasticity model. This is assuming the effects of water concentration, heterogeneous soil particles, and depth-dependent soil composition are negligible. The plastic behavior of granular materials such as soil, on the other hand, is more complex to model. As previously mentioned, there is extensive literature identifying the Drucker–Prager model as commonly used for representation of soil plasticity [53,54,55]. Drucker–Prager is a modification of the Mohr Coloumb [56] plasticity model and is a pressure-dependent model used to define the plastic yielding surface that depends on hydrostatic pressure and an associated flow rule.

The clay loam soil material model in this research was obtained from the paper by Naderi-Boldaji [52]. The mechanical properties that govern the soil behavior using the Drucker–Prager model are presented in Table 3. These material property values represent the type of soil that fire ants (Solenopsis invicta) typically use when building their nests.

### 2.5. Confined Compression Boundary Conditions and Interactions

The representative volume elements that were constructed and sized for their respective sections were prescribed a compressive loading as to assess their response to mechanical loading and to understand the role of the geometries/orientations of the cavities on collapse behavior. All nine RVEs were subjected to identical confined compression boundary conditions on lateral and bottom faces and uniformly enforced displacement loading at the top face. The applied displacement on the top face corresponded to a 20% global strain. This was chosen as it is representative of large deformations to ensure collapse while also being low enough to prevent the onset of excessive element distortion issues. The sides of the RVEs were laterally constrained on the faces, such that the x and y displacements were set to zero on their respective faces. The bottoms of the RVEs were fixed. This confined compression loading of the RVE models is illustrated in Figure 5.

In addition to the boundary conditions, additional interactions needed to be defined specifically related to the contact the RVEs’ internal cavity surfaces experience with themselves during collapse. The contact expected from the collapse of RVEs is defined as self-contact or contact of a body or surface with itself. At the RVE scale, this contact is significant enough to affect the global and local responses, thus requiring the definition of contact in the solution. It should be noted that the explicit dynamic solver was used for this analysis.

### 2.6. Mesh

The RVEs were meshed using linear tetrahedral (C3D4) elements. A uniform mesh of nominal seeding size in the ranges of [0.75–1] was used. The element size varied slightly between models due to attempts to resolve distorted elements. The fine mesh size was chosen so that it could accurately model the irregular cavity surfaces and capture the local stress buildup and plastic deformations. This was performed to prevent further stress concentrations from arising around the pore openings, which are more highly seeded in default meshing methods. The meshing details for all nine RVEs are listed in Table 4. A mesh convergence study was performed and reported in previous work [43].

### 2.7. Effective Elastic Properties

Homogenization is the process of obtaining effective macroscale properties of composite materials using numerical or analytical micromechanical models where the constituents are modeled explicitly. Tian et al. [57] demonstrated the use of homogenization using finite element methods and compared this with the standard two-step mean-field homogenization procedure.

In this work, the Micromechanics plugin tool developed for Abaqus™ software by Ross Mclendon is used [58]. The plugin offers an easy-to-use GUI interface that allows for the development of RVEs, applications of boundary conditions, and request outputs of effective properties. The loading cases needed to determine the effective properties are automatically generated by the plugin. Since the RVE analysis aspect of this thesis focuses on local geometries, micromechanical methods can be used to determine the effective elastic properties of the RVEs.

## 3. Results and Discussion

This section discusses the analysis of rectangular sub-volumes of the soil with ant nests that are considered to be representative volume elements for soils with ant nest cavities from three different depths of the full ant nest. In this section, the structural mechanisms are investigated by observing the loading response and corresponding effective elastic properties, and taking a more detailed look at the response in the plastic regime.

### 3.1. Loading Response

This section explores the stress vs. strain curves as well as the stress/displacement distributions to infer the differences in load carrying abilities. The three RVEs from each section are compared. Figure 6 presents the stress vs. strain responses of the RVEs in confined compression loading. It should be noted that the top RVE #3, which has a domain with a double aspect ratio, is the only model that does not reach a global strain level of 0.2. This is due to termination in the analysis that occurs because of excessive element distortion.

The stress–strain response of the RVEs are color coded for each group based on location: the RVE taken from the top in green, from the middle in blue, and from the bottom in red. The graphs show that the stress–strain behavior varies between the RVEs taken from the between the various regions. The stress–strain response of RVEs taken from the same regions exhibit small variations due to the randomness of the pore cavity structures in the RVEs. Despite these variations, the differences between RVEs taken from different regions are distinct and noticeable.

In the elastic response (lower stress and strain range), the elastic stiffness ranks high to low for RVE locations from bottom to top, respectively. The stiffness of the RVEs from the top and middle exhibits the least variation among the replicate RVEs from this level. However, the bottom RVEs exhibit large variances. The large variance in the stiffness of RVEs from the bottom is attributed to the fact that at the bottom of the nest, the cavities are fewer and larger, and these RVEs only incorporate a few of these cavities.

Most notably, the behavior in the plastic regime is different between RVEs from different regions. The primary differences between the response of the RVEs taken from the middle and bottom sections are the onset of plasticity. The pore geometries from the bottom section seem to resist loads much better and exhibit smaller changes in stiffness after yield. This is indicative of RVEs from the bottom being less susceptible to larger plastic deformations than those of the middle section. The RVEs taken from the top section have vastly different behavior than the RVEs taken from the middle and bottom regions and seem to show an almost elastic-plastic type of homogenized behavior, with the post-yield slope being very low.

The results suggest that the pore geometries from the bottom region have the most resistance to loading as well as resistance to further plastic deformation beyond yielding. This likely can be explained by the increased volume of solid material between the porous cavities seen for the bottom RVEs, which provide more load-bearing capacity. This allows for better protection against stress concentrations and strain localizations, which further allows these loads to be dispersed rather than allow for localized failure regions. The increased load-bearing capacity of the middle RVEs, with respect to the top RVEs, is attributed to the vertical orientation of the pores providing improved compressive load carrying than that of horizontally oriented pores.

The large plasticity and low collapse stress exhibited by the RVEs from the top region is likely caused by the dense and horizontal porous orientation with little solid soil spacing seen in the geometries of this region. The horizontal orientation causes these geometries to have lower stress-bearing capacity. Additionally, the lesser spacing between pores allows for many simultaneous localized failures in those areas due to accumulating stress concentrations and strain localizations. This might suggest that the top section serves to maximize collapse failure and provide energy absorption that protects lower ant nest chambers. On the other hand, the stress vs. strain behavior for the middle, and the bottom, appears to be a more damage-tolerant structure.

While pore volume fraction was expected to play a major role in the stiffness of the RVEs, it was found that it does not explain the differences in the compression response of RVEs. Although the RVEs’ sizes were chosen to keep pore volume fractions constant, due to the stochastic nature, it can only be done within a small range. For example, some RVEs taken from the bottom section have slightly higher value of pore volume fraction (i.e., 36.4%) compared to RVEs taken from the top (i.e., 34.7%) and middle (i.e., 31.3% and 33.4%) sections. Yet, the compression response of RVEs from the bottom demonstrates higher stiffness, higher stress to the onset of global yield, and resistance to plastic deformation. This suggests that the architectural geometry of ant nest cavities, cavity shape, orientation, branching lengths, and spacing play a significant role in determining the collapse response under compression, rather than the material properties. This finding is important for applications in real-world material design, as it can aid in the development of materials with high collapse resistance while maintaining low density.

To further illustrate these observations, contour plots of the displacement magnitude and von Mises stress for one RVE from each region analyzed in the confined compression are presented in Figure 7 and Figure 8. The contour plots of the displacement magnitude and von Mises stress for the remaining six RVEs can be found in previous work [43]. These displacement and stress contours correspond to a global compression strain of 20%.

For the RVEs from the top layer, the displacement shows a linear gradation through the thickness. The stress contours show the soil medium that bridges the cavities carry the loads and develop distributed stress concentrations. These bridging regions also correspond to regions undergoing plastic yield that contribute to the collapse of the porous structure. It can be seen in the middle RVEs that the cavity/pore structures have a combination of horizontal and vertical cavities. The collapse and compaction happen around the horizontal pores. This can be observed from the larger regions in the RVEs surrounding the vertical cavities undergoing a constant rigid body motion. The global deformation is a result of the yield deformations of the soil between the cavities. Since the middle region has large and more vertically aligned cavities, the regions with high stress values at yield are much fewer than those seen in the RVEs from the top region. The bottom RVEs display significantly large deformation across the solid soil regions between cavities at the center of the RVEs, with the top and bottom halves undergoing rigid body motion. However, in the von Mises stress contours, it can be observed that the material is more uniformly stressed through the solid soil regions in the bottom RVEs.

The comparison of plots above show that the porous geometry of these RVEs lead to non-homogenous displacement and stress distributions in the cross-section planes. The soil regions between the cavities form supporting structures to carry the load. It is also evident from the plots that local collapse due to stress concentrations form in the thin solid regions between the pores. The primary takeaway from these plots is that stress and displacement concentrations form in diagonal fashion around thin soil regions. This suggests that soil regions are under shear stress, and this is the primary mode of yielding in these support structures. This emphasizes the importance of the spacing between the porous chambers and size of the supporting soil regions between the cavities. These stress concentration bands are abundant in the top RVEs and less prevalent in the bottom RVEs. This is again likely due to the larger spacing between the pores and overall lobed geometry seen in the bottom-section RVEs.

### 3.2. Plastic Behavior

In order to further support the claims, in regard to the behavior after the onset of plasticity for the RVEs from different regions, this section presents contour plots of the plastic strain growth and the plastic strain dissipation work versus strain. A major goal of this section is to demonstrate that the porous structures of the ant nest in the different regions are tailored for different purposes in collapse resistance. For this, the work of Marhadi et al. [59,60,61] is drawn upon. In their work, they investigated the progressive failure of 2D trusses. Their work showed that the common assumption that redundant truss structures provide damage tolerance is not always true. The redundancy must be tailored. They separately optimized the redundant truss structures to maximize energy dissipated in damage and to maximize their load they can carry. The optimized structures were randomly perturbed to simulate damage. It was found that in redundant structures optimized for energy absorption, the redundancy leads to load redistributions that continue to propagate damage spatially and, by stable failure propagation, maximize the energy absorbed. However, in redundant structures optimized for peak load-carrying capacity, the redundancy leads to redirecting loads away from failing members, minimizes the damage growth, and provides damage-tolerant structures. 

Figure 9, Figure 10 and Figure 11 present the plastic strain growth exhibited in one RVE per region obtained from the confined compression analysis. The contour plots of the plastic strain growth for the remaining six RVEs can be found in previous work [43]. For the plastic strain growth contour plots, the iso surface contour option will be presented to best visualize how plastic strains grow specifically in the pores. The iso-surface contour displays areas where particular variable values intersect with the geometric model, forming surfaces where the variable maintains constant values throughout the domain. This type of contour visualization is advantageous to identify regions in the model that meet a specified critical value. Additionally, the magnitude of the plastic strains in the contour plots was filtered to a minimum of 0.01 to allow for the visualization of only the regions reaching plasticity. The growth in regions of plasticity is demonstrated using three frames for the global strain levels of 0.066, 0.13, and 0.2.

The plastic strain contour plots for the RVEs derived from the top region figures show that the top-section RVEs develop plastic strain localization in bands around thin solid material regions bridging the horizontally oriented pores. The contours show that plastic strains for the top RVEs occur at multiple places almost simultaneously at the onset of local plasticity and form bands around these thin bridging solid material areas. The failed regions then grow in these localized regions. This further supports the inference made in previous sections with respect to plastic yield via shear bands. This would suggest that this sectional geometry is utilized as an energy absorption mechanism, according to the criteria established by Marhadi et al. [59,60,61]. The plastic growth contours of the Middle RVEs show that they accumulate plastic strains in the vertical/helical porous chambers. The failed regions, rather than grow in magnitude, seem to diffuse through the depth of the chambers. This is likely due to the orientation of the chambers that allow it to have better load-bearing properties with respect to vertical compression loadings. This is indicative of a behavior referred to as compartmentalization by Kiakojouri et al. [62], in which introducing artificial discontinuities—either through physical changes or variations in structural properties such as stiffness and energy dissipation—designed to disrupt the structural behavior under extreme loading conditions, prevents damage from propagating and limits its spread following local failure. According to Mahardi et al. [59,60,61], this would indicate that the middle RVEs exhibit damage tolerance properties. This would be desirable for the ants as it maintains passageways from the bottom regions in the event of catastrophic collapse to the top-level porous chambers. The contour plots for the bottom RVEs indicate that the plastic strains develop in the thin solid soil areas that bridge the cavities; however, since the bottom RVEs are constituted of larger pores separated by large solid material regions, they are able to diffuse the loads into these thicker solid regions. The magnitude indicates that the pore geometries of the bottom region diffuse these strains and stresses through these thick regions more than the middle RVEs can through their solid soil regions bridging the cavities. This is again likely due to the increase of spacing between pores.

To further support these claims made regarding the plastic strains, contour plots of the yield stress were obtained and agreed with previously stated conclusions. These results can be found in previous work [43]. These plots further correspond with the yielded areas seen in the plastic strain regions. Additionally, the global plastic strain energy dissipation is plotted against the global strain illustrated in Figure 12.

Figure 12 plots the variation in the growth of the total plastic energy dissipated against the global strain for all nine RVEs in the RVE confined compression analysis to further explain the plastic behavior observations made in previous sections. The plot shows that the top RVEs demonstrate the highest level of total plastic energy dissipation and that the onset of the plastic energy dissipation occurs at the lowest level of global strain, about 0.05. The middle and bottom RVEs exhibit the onset of plastic dissipation at approximately the same strain value, about 0.07. However, the bottom RVEs exhibit a much lower dissipation rate overall. These plastic energy dissipation characteristics seen in the different RVEs demonstrate that failure, or plastic yielding, occurs at early stages and at a higher rate in the top RVEs (energy absorption) than the middle and bottom RVEs (damage tolerance). This supports the evidence that the top section of the ant nest geometry provides energy absorption as to shield the rest of the ant nest structure and that the lower two structures exhibit more damage-tolerant behavior. The bottom section’s porous structure does a better job of localizing the yield region due to its denser region of solid material. There is a clear distinction of behavior depicted in the graph by the top, middle, and bottom RVE, further elucidating the hierarchical behavior of the fire ant nest geometry due to variations in the region’s porous configurations.

### 3.3. Skeletonization of RVE Pore Geometry

To better understand the distribution and characteristics of porous geometries within different regions of the fire ant nest, it is helpful to examine the skeletonization of RVEs. Figure 13, Figure 14 and Figure 15 show the skeletonization of one RVE per region generated using VesselVio [46]. The skeletonization images for the remaining six RVEs can be found in previous work [43]. The skeletonization of the RVEs allows for better visualization of the porous geometries, but is not meant to provide any quantitative observations. The skeletonized volumes of the RVEs show that the spatial variation of the geometry of the pores in the RVEs from the top, middle, and bottom sections correlate with previously identified geometric trends seen in the full fire ant nest. It can also be observed that these trends remain constant in RVEs taken from the same region. The top RVEs are composed of many tightly packed horizontal pores, the middle RVEs are comprised of larger-volume vertical chambers with slightly increased pore spacing, and the bottom RVEs are configured with the largest volume of lobed pores with the highest value of pore spacing. These RVE skeletonizations serve as additional justification for the geometrical classifications of the top, middle, and bottom regions of the fire ant nest geometry.

### 3.4. Effective Properties

Using the micromechanics Abaqus™ plugin, developed by Ross Mclendon, the effective elastic engineering constants of the nine sectional RVEs were calculated and are presented in Table 5. By determining the effective elastic properties of the RVEs, the structural effects are quantified.

Table 5 shows that the top RVEs have the lowest values of elastic moduli. It can also be seen that the middle RVEs have the high values of elastic moduli in the vertical loading direction. On the other hand, the bottom RVEs demonstrate the highest overall values of elastic properties. Additionally, the RVEs from the bottom region have the highest variation in values of elastic properties. It can also be seen that the middle RVEs have relatively high value for the elastic moduli in the loading direction (3), with the exception of middle RVE 3. On the other hand, the top RVEs have the lowest value of the elastic moduli in the loading direction, thus highlighting the effect of the pore orientation with respect to loading. These observations further support previous arguments that the top porous orientation is designed for energy absorption, as it is more compliant and, thus, more susceptible to deformations and the effects of the vertical and helical pore orientation that allow for increasing loading bearing properties in the loading direction. Pore geometry in the bottom region is composed of highly stochastic lobed shapes. The trends depicted by these effective elastic properties support the behaviors exhibited in previous results. The elastic properties are also not entirely dependent upon the pore volume fraction, as RVEs from the bottom and middle with higher pore volume fractions have increased elastic properties when compared to some top RVEs with lower pore volume fractions. This highlights the idea that the behavior of these porous configurations is not entirely dependent on porous volume fraction, but rather, is more influenced by porous orientation and spacing.

Further examination of the effective moduli relations shows that most RVEs in all regions demonstrate highly isotropic behavior, despite the clear stochastic development of the pores. This again emphasizes the importance of the pore geometry and orientation on the hierarchical loading response of the RVEs, particularly in the plastic regime. Although these elastic properties appear to all have isotropic configurations, there are distinct collapse behaviors by region. The collapse response is mostly governed by the pore geometry and orientation.

## 4. Conclusions

This research presents an investigation of the structural designs and mechanisms exhibited in the pore geometries of fire ant nest cavities that allow for increased resistance to collapse through finite element analyses of models of reconstructed real fire ant nest pore geometries. To accurately reconstruct a model of the geometry from fire ant nest cavities, aluminum castings of fire ant nests were imaged using X-ray computed tomography and converted to solid models. To further explore the impact of regional pore geometry variation on the loading response, RVE analyses were completed on nine RVEs (three for each defined region through the depth).

It was observed that the variations of porous geometry in the three designated regions (top/middle/bottom) resulted in distinct responses to compressive loading. The top region geometries demonstrated the response with least stiff elastic response and the highest spatial growth of plastic strain in the RVEs. The bottom and middle regions illustrated much stiffer behavior; however, the bottom-region geometries illustrated the lowest spatial growth of regions undergoing plastic strain localization post-onset of plastic behavior. Behavior in the plastic regime was investigated to understand the differences in behavior after initial collapse. The pore architecture in the top portion of the nest appears to spread the damage and lead to higher energy absorption, thereby acting as a protective shield. The pore architecture of inner chambers in contrast localizes regions of high stress and plastic deformation and acts as structures designed for damage tolerance. Upon further observations of the geometries and effective properties with respect to the results of loading, it was concluded that the main mechanisms are associated with pore spacing and orientation. The dense horizontally oriented pores in the top region are more susceptible to large deformations and therefore energy absorption, while the large pore spacing and vertical orientation in the middle section provide more resistance to compressive loading. The larger pore spacing in the bottom section proves to provide extra strength in the plastic region as compared to that of the geometries found in the middle region. It is shown that the cavity geometry shapes, rather than the material properties, are the primary structural factors that influence the collapse response, more so than the pore volume fraction. These insights gathered from the findings could guide the design of lightweight and collapse-resistant cellular structures in various engineering applications.

The conclusions made from this research are as follows.

The pore geometry in fire ant nests demonstrates hierarchical behavior through the depth of the nest that can be characterized by its local geometry, which is dependent upon the region from which it is found: top, middle, or bottom.The geometry of the pores at the top of the fire ant nest portrays energy absorption responses as the stress concentration and strain localizations occur at various locations simultaneously and spread throughout. Alternatively, the lower two sections demonstrate damage tolerance behavior, defined by loads that are distributed locally, which can keep the region with stress concentrations and strain localizations minimal and localized.Additionally, pore volume fraction is not the primary design factor that allows for collapse resistance in fire ant nests; rather, it is the orientation and magnitude of spacing between pores.

Further research is required before applying these findings to practical engineering problems. A non-linear analysis of a full nest structure must be completed to elucidate the system-level response and how the interactions between the various porous geometries contribute to global collapse resistance. This is indicative of a resistance mechanism referred to as complexity by Kiakojouri et al. [62], in which systems where the overall behavior comes from how different parts of the system interact with each other, rather than just the sum of the individual parts. In these systems, the components follow local rules and influence each other in ways that are difficult to predict. Additionally, more full nests will need to be analyzed to establish statistically sufficient generalizations. Experimental compression tests upon ant nest geometries in soil material are also required for the validation of computational models.

## Figures and Tables

**Figure 1 biomimetics-09-00735-f001:**
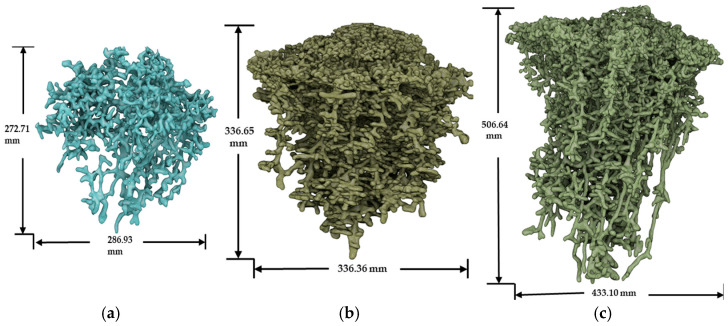
Visualization of ant nest casting: (**a**) Cast 0P1; (**b**) Cast 062; and (**c**) Cast 107.

**Figure 2 biomimetics-09-00735-f002:**
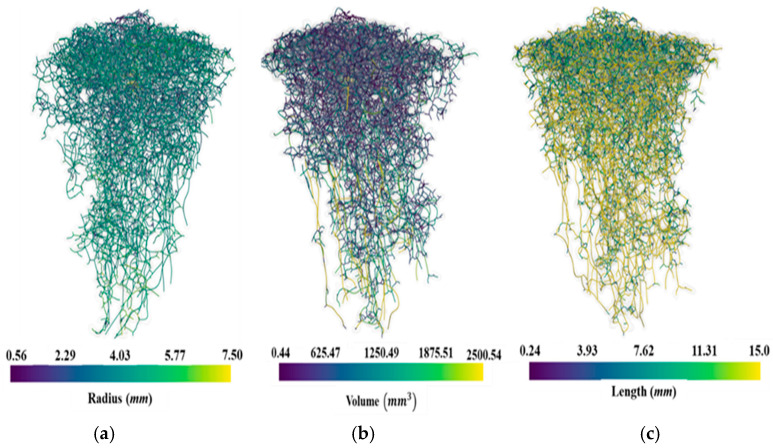
Skeletonized visualization of Cast 107: (**a**) radius, (**b**) volume, and (**c**) length.

**Figure 3 biomimetics-09-00735-f003:**
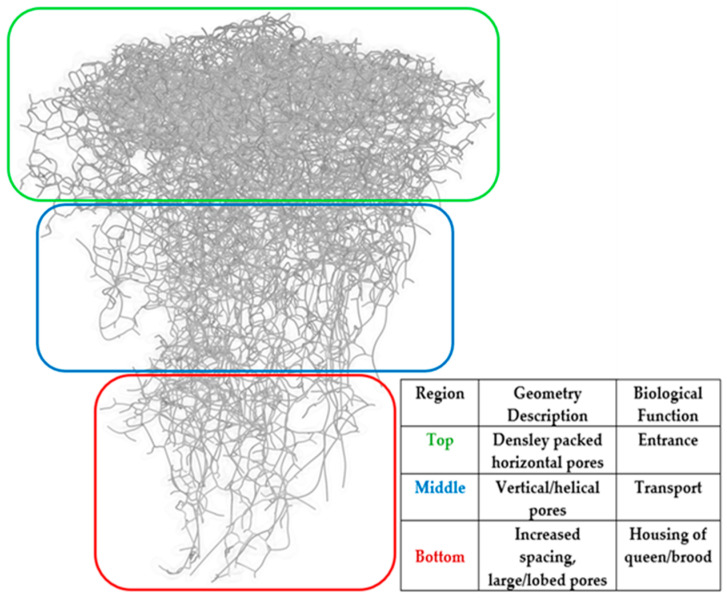
Fire ant nest RVE sectioning mapping.

**Figure 4 biomimetics-09-00735-f004:**
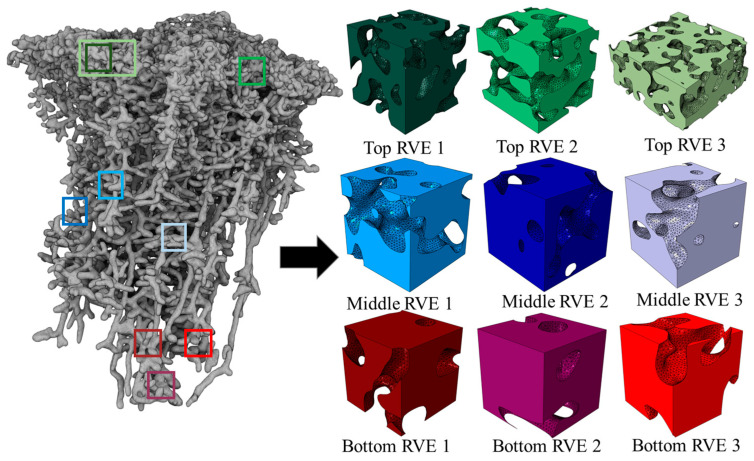
RVE sectioning location of fire ant nest cavity.

**Figure 5 biomimetics-09-00735-f005:**
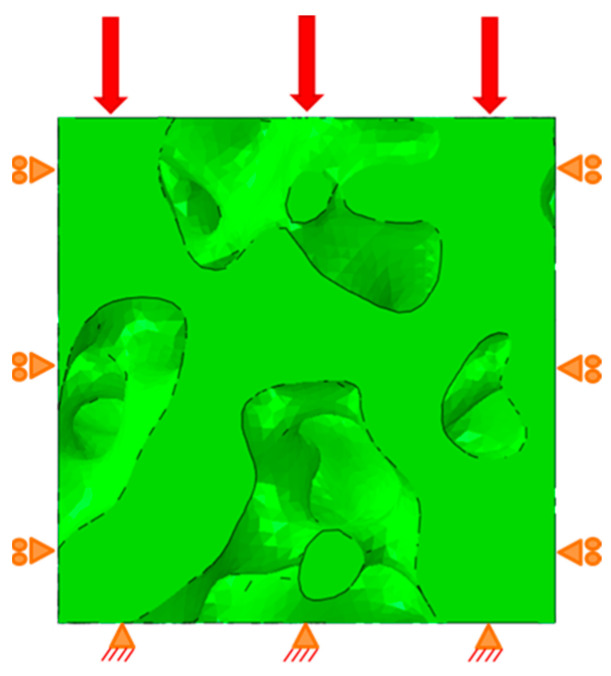
RVE confined compression boundary conditions.

**Figure 6 biomimetics-09-00735-f006:**
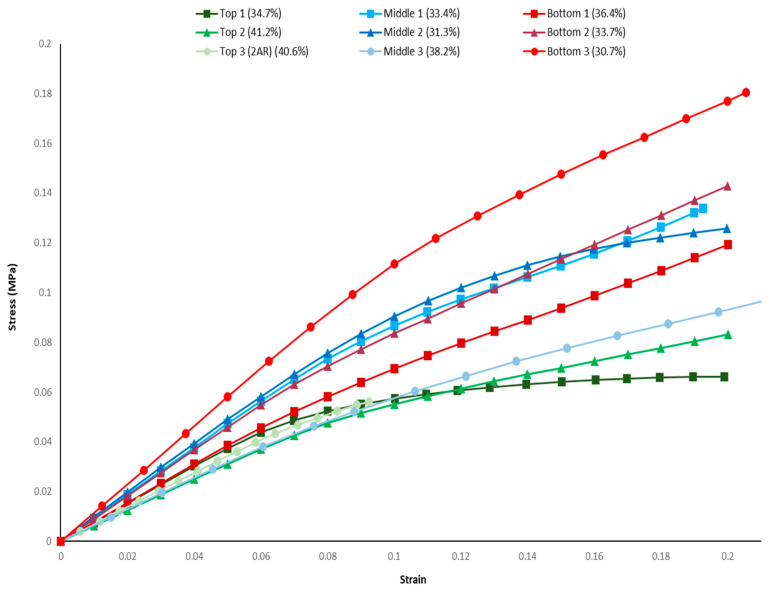
Stress vs. strain curve of regional RVEs.

**Figure 7 biomimetics-09-00735-f007:**
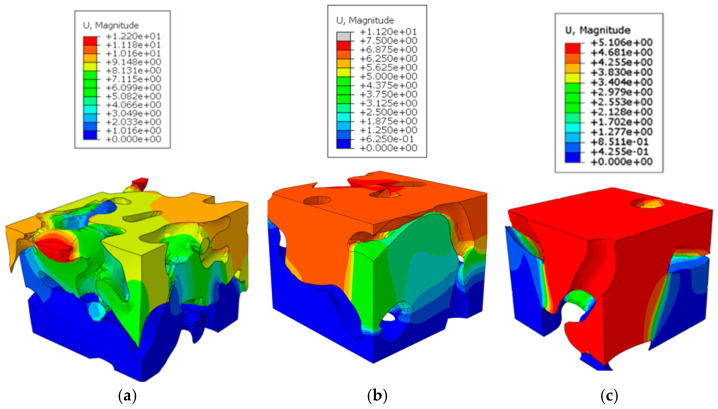
Displacement (mm) contours of RVE analysis indicating non-homogenous displacement distributions that lead to displacement concentrations forming in diagonal fashion around thin soil regions: (**a**) top; (**b**) middle; and (**c**) bottom.

**Figure 8 biomimetics-09-00735-f008:**
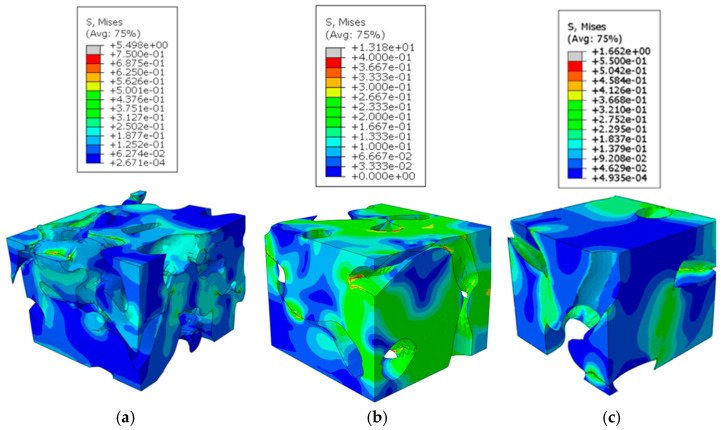
Von Mises (MPa) contours of RVE analysis indicating non-homogenous stress distributions around cavities that lead to local collapse: (**a**) top; (**b**) middle; and (**c**) bottom.

**Figure 9 biomimetics-09-00735-f009:**
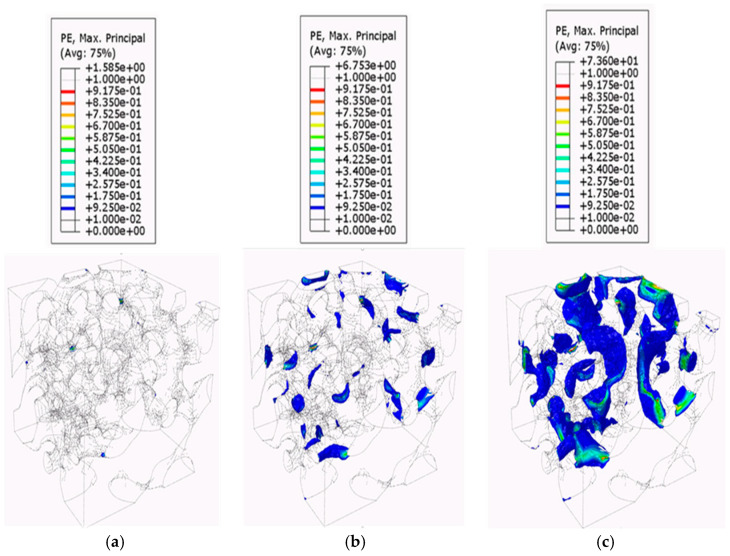
Plastic strain growth of the top RVEs: (**a**) 0.066; (**b**) 0.13; and (**c**) 0.2.

**Figure 10 biomimetics-09-00735-f010:**
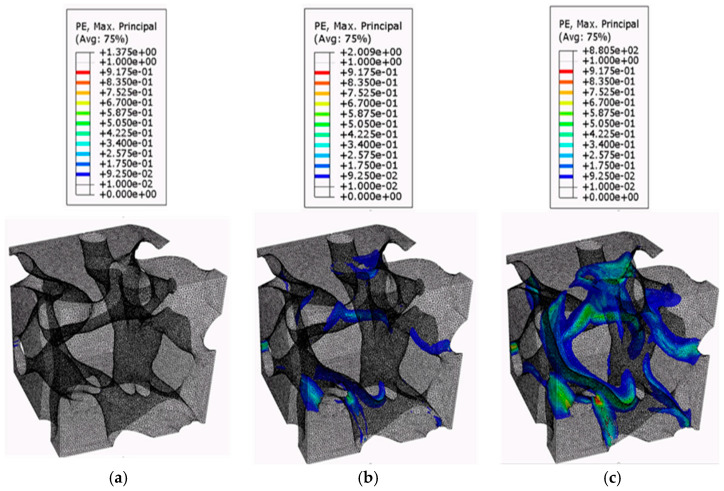
Plastic strain growth of the middle RVEs: (**a**) 0.066; (**b**) 0.13; and (**c**) 0.2.

**Figure 11 biomimetics-09-00735-f011:**
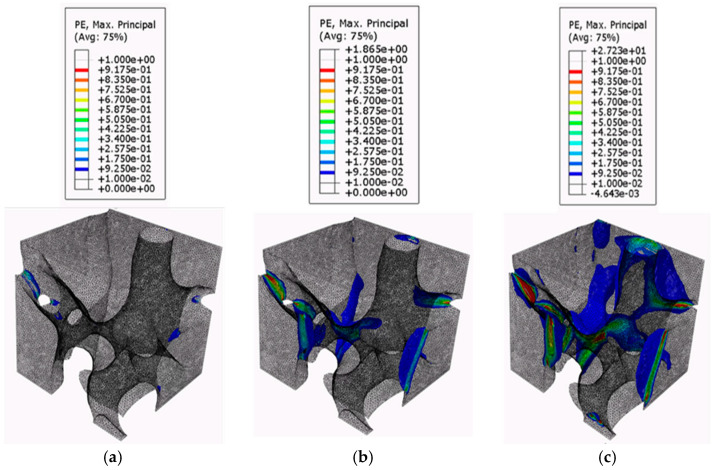
Plastic strain growth of the bottom RVEs: (**a**) 0.066; (**b**) 0.13; and (**c**) 0.2.

**Figure 12 biomimetics-09-00735-f012:**
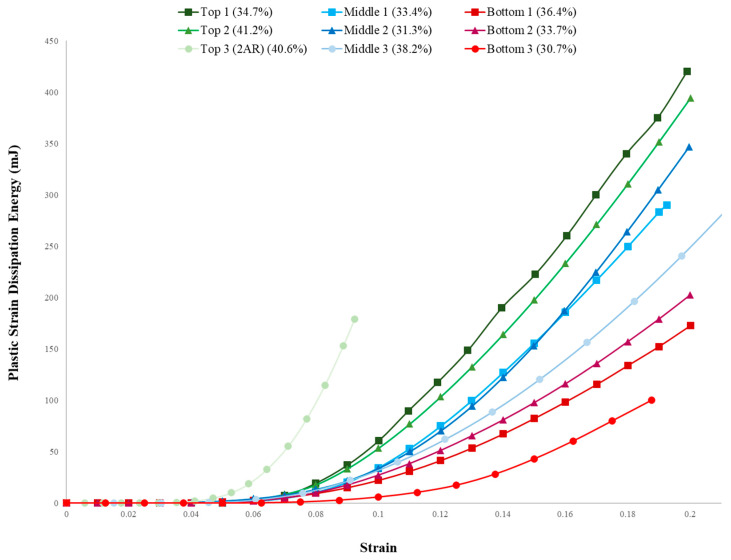
Plastic strain energy dissipation vs. strain.

**Figure 13 biomimetics-09-00735-f013:**
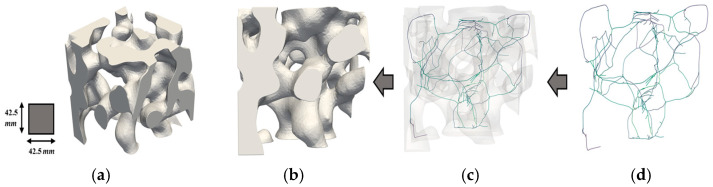
Skeletonization of Top RVE 1’s porous network: (**a**) isometric view; (**b**) original volume (**c**) superimposed volume; and (**d**) skeleton volume.

**Figure 14 biomimetics-09-00735-f014:**
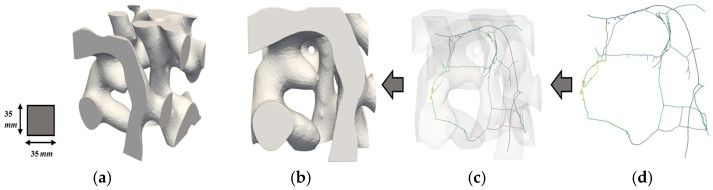
Skeletonization of Middle RVE 1’s porous network: (**a**) isometric view; (**b**) original volume (**c**) superimposed volume; and (**d**) skeleton volume.

**Figure 15 biomimetics-09-00735-f015:**
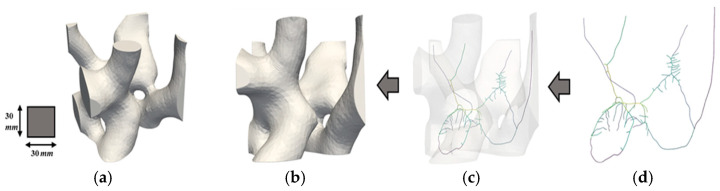
Skeletonization of Bottom RVE 1’s porous network: (**a**) isometric view; (**b**) original volume (**c**) superimposed volume; and (**d**) skeleton volume.

**Table 1 biomimetics-09-00735-t001:** Image processing parameters.

Image Processing Technique	Numerical Value
Grayscale Thresholding	0.1–3071
Smoothing (Closing)	2.5 mm

**Table 2 biomimetics-09-00735-t002:** RVE porous volume fraction.

RVE	Top RVE 1	Top RVE 2	Top RVE 3	Middle RVE 1	Middle RVE 2	Middle RVE 3	Bottom RVE 1	Bottom RVE 2	Bottom RVE 3
Pore Volume Fraction	34.7%	41.2%	40.6%	33.4%	31.3%	38.2%	36.4%	33.7%	30.7%

**Table 3 biomimetics-09-00735-t003:** Finite element material properties.

Material Behaviors	Material Parameters	Numerical Value
Elastic	Young’s Modulus (MPa)	2
	Poisson’s Ratio	0.3
	Density (kg/${mm}^{3}$)	1.52 × 10^−9^
Drucker–Prager	Angle of Friction	22°
	Flow Stress Ratio	0
	Dilation Angle	22°
Drucker–Prager Hardening	Yield Stress (MPa)	0.2
	Absolute Plastic Strain	0

**Table 4 biomimetics-09-00735-t004:** RVE mesh element counts.

RVE	1	2	3
Top	988,236	890,263	1,767,996
Middle	210,992	246,130	760,365
Bottom	773,536	646,743	332,615

**Table 5 biomimetics-09-00735-t005:** Effective elastic engineering constants.

Region	RVE	Elastic Moduli (MPa)			Poisson’s Ratio			Shear Moduli (MPa)		
		$E_{1}$	$E_{2}$	$E_{3}$	$v_{12}$	$v_{13}$	$v_{23}$	$G_{12}$	$G_{13}$	$G_{23}$
**Top**	1 (34.7%)	8.89	8.96	8.81	0.24	0.25	0.24	3.56	3.63	3.55
	2 (41.2%)	7.89	7.72	6.84	0.23	0.25	0.23	3.10	2.93	2.75
	3 (40.6%)	7.10	7.18	6.91	0.23	0.24	0.23	2.85	2.84	2.85
	Avg.	7.96	7.95	7.52	0.23	0.25	0.23	3.17	3.13	3.05
**Middle**	1 (33.4%)	9.54	9.61	10.02	0.25	0.24	0.24	3.79	3.84	3.87
	2 (31.3%)	10.97	9.52	10.33	0.25	0.24	0.24	4.10	4.24	3.90
	3 (38.2%)	9.22	9.46	9.21	0.25	0.26	0.29	3.68	3.15	3.25
	Avg.	9.91	9.53	9.85	0.25	0.25	0.26	3.86	3.74	3.67
**Bottom**	1 (36.4%)	9.64	9.40	10.97	0.26	0.24	0.24	3.79	3.64	3.63
	2 (33.7%)	9.64	12.14	10.81	0.25	0.25	0.27	4.52	4.23	4.43
	3 (30.7%)	10.98	8.54	10.04	0.27	0.26	0.22	3.89	4.11	3.83
	Avg.	10.09	10.03	10.61	0.26	0.25	0.24	4.07	3.99	3.96

## Data Availability

The original contributions presented in the study are included in the article, further inquiries can be directed to the corresponding authors.

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
