# Peer review of "Elucidating Collapse-Resistant Mechanisms of Pore Geometries in Fire Ant Nest Cavities"

_biomimetics, 2024, doi:10.3390/biomimetics9120735_

Round 1

Reviewer 1 Report

Comments and Suggestions for Authors

The manuscript presents a study on the collapse resistance mechanisms of pore geometries in fire ant nests. Finite Element Models of ant nests in soil were developed using X-ray CT imaging of aluminum castings. The findings suggest that the pore architecture in the upper portion of the nest disperses damage and increases energy absorption, while the architecture in the lower chambers localizes high-stress regions and plastic deformation, demonstrating damage tolerance. The study is highly relevant and fits within the journal's scope. While the manuscript is generally well-organized, the following points need to be addressed before consideration for publication.

1.     While the final decision rests with the authors, I see no justification for the subsections in Section 1. Additionally, this section is overly long and lacks focus on the current study.

2.     I believe there is novelty in this work, but it is not sufficiently emphasized in the manuscript. Please highlight the research gap and the main contribution of this study in Section 1.

3.     Ensure compliance with the journal’s policy regarding the number of keywords.

4.     Please ensure that SI units are used consistently throughout the manuscript (for temperature for example). If the authors prefer to include other units, they can do so by adding them in parentheses after the corresponding SI unit.

5.     Be meticulous with capitalization when referencing tables and figures in the text.

6.     There are also a few typesetting issues in the manuscript. Please review and correct them.

7.     While the manuscript offers a structured and comprehensive literature review, there is still room for improvement. I suggest consulting this recent review article (https://doi.org/10.3390/biomimetics9090545), particularly its classification.

8.     It is unclear whether the authors scanned only three nests. If so, do they believe this number is statistically sufficient for generalization? Or, were more nests investigated, and these three are just examples of possible classes?

9.     For commercial products and software, please provide reference links where possible. Be accurate with naming conventions. For example, "ABAQUS" should be corrected to "Abaqus."

10.  There are a few well-known resistance mechanisms in biological structures under extreme loading conditions. For example, in this work (https://doi.org/10.3390/biomimetics8010095), complexity and compartmentalization techniques are emphasized. I think they are also recognizable in fire ant nest. It is recommended to consult this research and enhance the discussion accordingly.

11.  It is suggested to include a figure (or updating one of the existing ones) showing the nest's utilization, with particular emphasis on highlighting the queen's region.

12.  Referring to the loading applied in the FE model, more background and justification are needed. What is the natural extreme load experienced in real nests? This aspect requires further discussion.

13.  The manuscript is commendable for its clarity in methodology and its thorough detailing of the techniques used. I greatly appreciate this, as ensuring reproducibility is a crucial but sometimes overlooked aspect of scientific research. However, a few points need clarification, such as which Abaqus solver was used and if any special solver configurations were applied in the numerical study.

14.  In all the contour figures, such as those reported in Figures 7 and 8, the units should be specified in the captions; otherwise, the interpretation of the results is impossible.

15.  The different regions of the nest are discussed, highlighting the varied behaviors in these areas. However, the boundaries of these regions are not clearly explained. Is there a transient region between these three areas? Is it distinct in all natural nests? More discussion is needed in this regard.

16.  This reviewer wonders why the system-level response is overlooked. I believe the authors have already developed the necessary tools and materials for a finite element study of a complete nest, which could provide broader insights into the system's global collapse behavior. The authors could suggest this as a future research need.

Reviewer 2 Report

Comments and Suggestions for Authors

Abstract (Lines 10-24):

  • Line 12: The sentence, “biological adaptations are likely to have evolved these structures to have an increased ability to resist collapse,” feels a bit convoluted. I suggest rewording it for clarity. Consider something like, "These structures may have evolved biological adaptations that enhance their collapse resistance."
  • Lines 20-21: The description of the pore architecture at different depths is well-explained, but it’s a bit dense. Breaking this into two sentences might improve readability. For example, “The top portion acts as a protective shield by distributing damage and absorbing energy. In contrast, the lower chambers localize stress, contributing to damage tolerance.”
  • Line 23: A final sentence summarizing the practical implications of the study would help make the abstract more impactful. You could add something like, "These findings could inform the design of lightweight and durable cellular structures in various engineering fields."

Introduction (Lines 27-67):

  • Line 35-37: The sentence discussing the need for vented materials in space could be more concise. Instead of, “In general, open cell cellular materials or vented cellular materials typically have lower stiffness and strength than closed cell cellular materials,” you could say, "Vented cellular materials generally have lower stiffness and strength than closed-cell structures."
  • Lines 44-50: You mention the potential of additive manufacturing for replicating complex porous structures but don't address the limitations or challenges. Briefly touching on these challenges (e.g., material limitations, cost) would add depth to the discussion.
  • Lines 60-67: Your hypothesis is strong, but you could emphasize the novelty of your approach by adding something like, “This study uniquely applies fire ant nest architectures to engineered structures, a largely unexplored area of biomimetic design.” This would highlight how your research stands apart from others.

Materials and Methods (Lines 180-264):

  • Line 185: When introducing the fire ant nests (Solenopsis Invicta), it would be helpful to explain why these specific nests were chosen. For instance, do they exhibit particularly interesting architectural features that make them ideal for this study? Providing a rationale would add context.
  • Line 211: More information on the X-ray CT scan resolution would be beneficial. Specifically, what impact does the resolution have on your ability to accurately model the nests? Were there any limitations in capturing fine details that might influence your results?
  • Line 221: While 3-D Slicer is a commonly used software, a brief explanation of why this tool was chosen for your study (versus other available software) would strengthen your methodology section.

Results and Discussion (Lines 434-639):

  • Line 454: You mention that the stress-strain response varies between the top, middle, and bottom RVE, but the explanation could be more detailed. For example, what physical factors are causing the top region to have less stiffness? Expanding on the reasoning behind these mechanical behaviors would make your discussion more robust.
  • Line 470-478: While you state that the bottom section resists loads better due to larger solid regions between pores, it would be useful to explain whether this is a result of material properties, pore geometry, or a combination of both. Delving into the mechanics behind this resistance would enhance the readers' understanding.
  • Line 499-501: You make a key point that pore geometry plays a bigger role in collapse resistance than pore volume fraction. This is an important finding, but it could be expanded to discuss practical applications. For instance, how can this insight be utilized in real-world material design? Expanding on potential applications would greatly enhance the relevance of your findings.

Figures and Tables:

  • Figures 7 and 8: The von Mises stress and displacement contour plots are clear, but the figure captions could be more descriptive. For instance, explaining what the reader should specifically notice in the stress distributions would help those less familiar with FEA results interpret the data more effectively.
  • Table 5: The elastic constants are well-presented, but adding a comparison to other porous materials or previously studied biomimetic structures would provide context and emphasize the importance of your findings.

Conclusion (Lines 716-757):

  • Line 744-745: The conclusion that pore geometry is the primary factor in collapse resistance is strong, but this section would benefit from a statement about future directions. What are the next steps in applying these findings to practical engineering problems? Mentioning future research or potential industries that could benefit would make your conclusion more impactful.
  • Lines 746-757: The conclusion is well-written, but it could be more actionable. For example, you could add: “These insights can guide the design of collapse-resistant structures in aerospace, civil engineering, and beyond.”This would help highlight the real-world implications of your research.

Reviewer 3 Report

Comments and Suggestions for Authors

1、 There are so many keywords that could confuse the focus of this work. I suggest the authors to concentrate the main findings and provide 5 keywords at most.

2、 The authors should provide adequate evidences or references to support the describing on the ant nest architectures (Tshnikel) (line 52-57).

3、 The logical relationship among “cellular materials”, “biomimetic design” and “aerospace application” in Section 1.2 is not clear. I suggest the authors to re-organize this section with more recent related literature on biomimetic design of lightweight structures, such as Zhang et al. Biomimetics 2023, 8(2), 153; Chen et al. Biomimetics 2023, 8(2), 140; Zhong et al. Biomimetics 2023, 8(3), 284.

4、 Apart from the unique architecture of the fire ant nests, the nests’ material properties could also play a key role in its excellent mechanical performance. The authors should provide adequate explanation why the aluminum was selected for the nest model rather than any other similar materials.

5、 Although this work has provided sufficient numerical simulation results to elucidate the possible underlying collapse resistant mechanism of the fire ant nest, real experimental results are much more convincing. I suggest the authors to provide necessary mechanical experimental data in the revised version.

6、 The authors also should double check the text to ensure the normative format of the manuscript, such as the punctuation marks (line 73), case sensitivity (line 79, 106, line 199) and so on.

Round 2

Reviewer 1 Report

Comments and Suggestions for Authors

The authors have addressed my main concerns, and the manuscript's quality has improved significantly. I am now inclined to accept this manuscript for publication. However, there are still a few remaining issues that need attention.

1.     There are some problems in in the references list, I am pointing out a few examples here, but the entire list should be carefully double-checked for consistency and accuracy:

-In ref. [63], please check and update the standard format for a conference paper accordingly.

-For all journal papers, please remove the publication month, as it is not commonly included in standard referencing styles.

-Ref. [64] appears inconsistent with the discussion in the main text; please update to ensure clarity and relevance.

-Several entries are missing necessary details, such as the article number in ref. [34].

-There are multiple instances of incorrect punctuation that should be reviewed and corrected.

2.     Additionally, please remove the capitalization of the first letter from the captions of tables and figures, as it does not align with the style used by Biomimetics.

3.     The term “collapse mechanisms” is listed as a keyword in the manuscript, but it has not been used anywhere in the text. Please either remove it, replace it with a more suitable phrase, or incorporate it into the manuscript where appropriate.

Reviewer 3 Report

Comments and Suggestions for Authors

The authors have addressed all my concerns.